# Pesticides: Unintended Impact on the Hidden World of Gut Microbiota

**DOI:** 10.3390/metabo14030155

**Published:** 2024-03-07

**Authors:** Asghar Ali, Khalid I. AlHussaini

**Affiliations:** 1Clinical Biochemistry Laboratory, Department of Biochemistry, School of Chemical and Life Sciences (SCLS), Jamia Hamdard, New Delhi 110062, India; 2Department of Internal Medicine, College of Medicine, Imam Mohammad Ibn Saud Islamic University (IMSIU), Riyadh 4233-13317, Saudi Arabia; kialhussaini@imamu.edu.sa

**Keywords:** persistent organic pollutants, dysbiosis, metabolic homeostasis, organochlorine organophosphate pesticides, heavy metals, health hazards, insecticides, short-chain fatty acids

## Abstract

A vast range of pesticides have been routinely employed for plant protection throughout the last few decades. Pesticides can enter non-target organisms in various ways, posing health hazards. Exposure to different environmental pollutants, including pesticides, can affect the human gut flora. Metabolites generated from the gut microbiota play an essential role in the host’s health by regulating metabolic homeostasis. A disruption in this equilibrium can lead to the emergence of numerous illnesses and their etiology. Pesticides have been shown in a few recent studies to harm the host’s gut microbiome. As a result, there is an urgent need to investigate the impact of pesticides on gut microbiota-mediated immunity. Metabolic alterations in the host may give a better understanding of pesticide-induced harm. This review highlights the potential consequences of pesticide exposure on gut microbiota composition and function, mainly focusing on how it might alter the production of secondary metabolites with potential downstream implications for host health.

## 1. Introduction

Food consumption has grown exponentially as the world population is rising. Pesticides of various varieties are now extensively employed worldwide to improve agricultural product quality and enhance crop yields, resulting in major economic benefits. Pesticides enter the soil, water, air, and non-target creatures such as people [1]. Various medical problems concerning pesticide hazards to animals have risen dramatically [2]. A growing number of studies have linked pesticides to a variety of pathologies, including, immune system dysregulation, neurotoxicity metabolic diseases (such as obesity and type 2 diabetes), endocrine alterations, reproductive disorders, and even tumors, whereas the gastrointestinal microbiota plays a critical role in a variety of host metabolic and immune functions [3]. In most circumstances, the gut can come into direct contact with dietary pollutants, and pesticides are likely to be absorbed by the human gastrointestinal tract and gut flora. Consider an in vitro study that found that exposing bacterial colonies directly to the herbicide glyphosate might modify the constitution of the bacterium [4]. Many authors have specifically proven that different types of environmental contaminants, including certain pesticides, may impact and modify gut microbiota (GM), hence the function of the gut microbiota in pesticide-induced toxicity in non-target species is receiving interest [5]. This review aims to provide an overview of the relationship between exposure to pesticides, changes in the gut microbiota, and the effects of their secondary metabolites on the health of hosts, as well as the mechanism(s) by which various persistent organic pollutants cause changes in the composition and function of the microbiota.

This review explores the intricate relationship between pesticides and the gut microbiota, delving into the unintended consequences of pesticide exposure on the composition and functionality of these microbial communities. We scrutinize recent findings, highlighting the potential disruptions caused by pesticides to the delicate balance of the gut microbiota, influencing its diversity and metabolic activities. Additionally, we assess the role of different classes of pesticides in shaping the microbiota and explore the mechanisms by which these xenobiotics may exert their effects. 

## 2. Gut Microbiome 

Gut microbiota are microbes that dwell in the gastrointestinal (GI) tract and constitute the gut microbiome when coupled with their bioactivity and genetic material. The human gastrointestinal (GI) tract contains billions of diversified microorganisms, including viruses, bacteria, fungi, and protozoa. The human gut is dominated by mainly four phyla: Proteobacteria, Firmicutes (*Clostridium, Lactobacillus*, *Eubacteria*, and *Peptoniphilus*), Actinobacteria (*Bifidobacterium*), and Bacteroidetes (*Prevotella*, *Bacteroides*) (Table 1). The Firmicutes and Bacteroidetes phyla account for around 90% of all microorganisms, each having a distinct role [6]. *Methanosphaera stadtmanae* and *Methanobrevibacter smithii* are examples of Archaea, whereas yeast predominates the Eukarya in the gut [7]. However, various individual differences and underlying variables such as food, pregnancy, hormonal changes, age, illness, gender, and medication (e.g., antibiotics and proton pump inhibitors) result in variations in the composition and concentration of gut microbes [8]. Gut microbes play a significant role in regulating metabolism (e.g., bile acids) and body homeostasis by controlling immune, digestive, and neurological processes with the help of the gut–brain axis (GBA–a network of highly interrelated complicated physiological pathways). In response, the human host provides a nutrient-rich environment [9]. Other functions of gut microbes in children are the strengthening of immunity of the gut mucosa (e.g., induction of secretory regulatory T cells and secretory IgA) and structural gastrointestinal development (e.g., payer’s patches–gut-associated lymphoid tissues, epithelium) [10]. Many proteins which are not present in the human body are encoded by the gut microbes, e.g., enzymes required for the breakdown of hemicellulose and indigestible dietary fibers [9]. Gut bacteria synthesize short-chain fatty acids and monosaccharides by fermenting nutritional fibers. SCFAs (short-chain fatty acids) like acetate and butyrate are necessary energy providers [11]. 

### 2.1. Function of Gut Microbe Metabolites

The human host and the gut microbiota interact as a “metabolic organ” which carries out a variety of vital tasks to safeguard human health [17]. The human host can make use of various energy sources due to the gut bacteria’s metabolic activities. Studies on the metabolic profiles of humans and animals have shown that the microbiota can particularly influence how dietary lipids are absorbed, stored, and used. Understanding the functions of these gut microbiome metabolites is crucial for unravelling their impact on host health. The gut microbiota produces organic fatty acids called SCFAs (short-chain fatty acids) through the anaerobic fermentation of non-digestible proteins and fibers, mostly from acetate, propionate, and butyrate [18]. SCFAs play a crucial role in energy metabolism, immune modulation, and maintaining gut barrier integrity. Recent studies have highlighted their potential therapeutic applications in conditions such as inflammatory bowel disease and metabolic disorders [19]. The production of SCFAs is a crucial “microbiota function”. Intestinal epithelial cell proliferation and differentiation are positively influenced by SCFAs, particularly acetate, propionate, and butyrate, which also have diverse metabolic properties [20]. Actually, under normal circumstances, SCFAs supply 5–10% of the usual energy requirements. Key microbial species involved in this process, such as *Bacteroides* and Firmicutes, contribute to the enzymatic breakdown of complex carbohydrates, yielding SCFAs [21]. The colonic defense barrier can be strengthened by butyrate by increasing mucin production, trefoil factors, and antimicrobial peptides. SCFAs influence host energy metabolism by serving as substrates for mitochondrial beta-oxidation. Additionally, they enhance gut barrier integrity through the upregulation of tight junction proteins, such as ZO-1 and occludin [22]. Butyrate is employed as an energy source for colonocytes. Additionally, butyrate has several impacts on a range of cell types, including immunological regulation, cell cycle inhibition, the induction of programmed cell death, and cellular differentiation [14]. Gut microbiome metabolites also include bile acids, which are produced through the metabolism of host-derived bile. The gut microbiome influences bile acid composition, impacting host metabolism and signaling pathways [23]. The GM modulates bile acid homeostasis. By modulating the expression of bile acid-production enzymes, the microbiome participates in the synthesis of principal bile acids, cholic acid and chenodeoxycholic acid. Other bile acid metabolism processes affected by the GM include conjugation in the liver, reabsorption in the terminal ileum, and deconjugation in the small intestine [24,25]. Figure 1 summarizes the role of metabolites.

### 2.2. Dysbiosis

Gut dysbiosis refers to disbalance in the function and composition of gut bacteria, alteration in the distribution of gut bacteria, or changes in bacterial metabolic activities. Gut dysbiosis can result from many factors, including antibiotic use, dietary choices, lifestyle, and genetic predispositions [26]. Though antibiotics effectively target pathogens, they can unintentionally disrupt the equilibrium of beneficial microbes, resulting in dysbiosis. Additionally, consuming diets rich in processed foods and deficient in fiber fosters an environment that promotes dysbiosis. Lifestyle factors, such as stress and lack of physical activity, further contribute to microbial imbalance [27]. Studies have revealed that dysbiosis is often characterized by a reduction in beneficial bacteria, such as *Bifidobacterium* and *Lactobacillus*, coupled with an overgrowth of potentially harmful microbes, including certain species of *Clostridium* and *Enterobacteriaceae* [28]. This shift in microbial composition can profoundly affect the host’s health, influencing immune function, nutrient absorption, and overall gut homeostasis. The disruption of microbial balance leads to alterations in immune function, increased intestinal permeability, and systemic inflammation [29]. Dysbiosis comprises three types: (i) Decline in diversity of gut microbes. (ii) Reduction in valuable organisms. (iii) Proliferative growth of possibly hazardous microbes [30]. Two primary variables might contribute to dysbiosis: host-related factors and environmental factors [31]. Symptoms might initially point to dysbiosis in a person before a particular health issue is detected. Vaginal or rectal itching, foul breath (halitosis), diarrhea, constipation, bloating, nausea, and chest discomfort are typical dysbiosis symptoms. Dysbiosis can also cause malnutrition due to a disturbed digestive system. Depression, reduced concentration level, and anxiety are also possible cognitive weakening symptoms of dysbiosis [32]. Considering the vital role of the gut microbiota in maintaining the homeostasis of the host, its influence spans across all the organ systems of the human body, expanding to various extraintestinal sites. Such bidirectional influences are extended via the gut–brain axis, the gut–heart axis, the gut–liver axis, the gut–lung axis, the gut–skin axis, and the gut–reproductive axis, among others. It has been proved that dysbiosis is one of the leading causes of many chronic diseases like obesity, type 1 and 2 diabetes, gastrointestinal cancer, inflammatory bowel disease, non–alcoholic fatty liver disease (NAFLD), chronic kidney disease (CKD), cardiovascular diseases such as acute ischemic stroke (AIS), and it was even suggested to play a role in the development and progression of neurological disorders such as Alzheimer’s disease [33].

## 3. Effect of Insecticides on Gut Microbiota

Insecticides, a crucial factor in increasing agricultural output, are used to kill problematic insects. However, these pesticides often exhibit off-target effects due to prolonged exposure [5]. According to recent research, they are hazardous to non-target creatures in the environment. Furthermore, the amount or concentration of insecticides increases as they go up the food chain, posing a concern to human health. Multiple studies have recently discovered pesticides in the human body [34]. Once inside the human body, traces of pesticides may also exhibit adverse effects on the gut microbiota, which can negatively impact human health and behavior [35]. Various exposure studies in this regard have shown that exposure to xenobiotics such as pesticides can lead to alterations in the gut microbial composition, leading to dysbiosis, which further leads to altered functions and behavior impairments in the host, as demonstrated by using animal models [36]. These changes in the gut microbiota can result in various health effects, including metabolic and endocrinal disorders, dysregulation of the immune system, inflammations, and impact on the gut–brain–blood axis as well [37]. Given the extensive employment of pesticides in modern agricultural practices, and the active or passive responses of gut microbiota to such xenobiotics, broader insights must be gained with regard to the interaction between the gut microbiome and pesticides and their long-term collateral effects on gut microbiota composition and function. In the following sections, the effects of various commonly used insecticides on gut microbes are elucidated.

### 3.1. Organophosphate Pesticides (OPPs)

Amides and phosphoric acid are the sources of this class of insecticides. OPPs are the most widely used pesticide worldwide because of their biodegradable nature. These chemicals’ remnants are present in the air, on plant surfaces, and in soil. These chemicals leach into the human system through water and soil. OPP residues have been discovered to be strongly correlated with diabetes in samples of human plasma. When gut microbes break down OPP, the esterase activity and acetate are changed, which leads to gluconeogenesis and glucose intolerance [38]. According to a prior study, mice exposed to OPPs for 180 days developed glucose intolerance. This condition is brought on by the gut microbiome’s breakdown of organophosphate into acetic acid, which is then used as a substrate for gluconeogenesis [39].

Frequently found in fruits and vegetables, chlorpyrifos (O, O-diethyl, O-(3,5,6-trichloro-2-pyridyl)-phosphorothioate) is a widely used organophosphorus insecticide. Several earlier research studies have shown the effects of CPF (chlorpyrifos) in both in vitro and in vivo settings. In the SHIME^®^ model, which mimics the intestinal environment in vitro, there was an observed decrease in the levels of beneficial bacteria such as *Bifidobacterium* and *Lactobacillus*, along with an increase in the abundance of *Enterococcus* and *Bacteroides*. This shift in microbial composition may lead to alterations in pH levels or the stimulation of short-chain fatty acids (SCFAs), which in turn can inhibit the colonization of potentially harmful bacteria in the gut [40]. When CPF is applied alone to Caco-2/TC7 cell models of the intestinal mucosa, this can disrupt the mucosal barrier, cause the release of the chemokine IL-8, and cause inflammation. However, as mentioned earlier, the two models do not account for microbiota–host crosstalk [41]. Prolonged exposure appears to have effects on the gut in addition to the levels of the gonadotropin’s follicle-stimulating hormone, luteinizing hormone, and testosterone in the blood, as well as the inflammatory cytokines IL-6, monocyte chemoattractant protein-1, and TNF- in rats, suggesting a potential role in the development of colitis and infertility [42]. The effects of CPF on oxidative stress and gut microbiota dysbiosis in zebrafish have been emphasized by other scientists. In comparison to the control group, researchers found that zebrafish exposed to CPF exhibited an increase in Proteobacteria abundance alongside a decrease in the Bacteroidetes phylum [43]. In contrast to mice treated for 30 days with 1 mg/kg body weight of CPF, which caused changes in the gut microbiota and urine metabolites, it was found that amino acids, metabolites, SCFAs, and bile acids caused intestinal inflammation and aberrant permeability of the intestine [44]. Last but not least, exposure to CPF reduced hepatic glucose and lipid metabolism in an animal model through oxidative stress and microbiota dysbiosis [44].

Due to its widespread use in agriculture, diazinon is another OPP that has caused public health issues. Drinking water, the main method of exposure to diazinon for people, has a substantial traceability in its amount [45,46]. Male mice exposed to 4 ppm diazinon concentrations in drinking water for 13 weeks showed changes in bacterial populations and composition, as well as significant impairments in the energy metabolism of the gut flora. Additionally, exposure to diazinon activated several stress response pathways [46]. Rats exposed to diazinon for 14 days exhibited histological alterations and an upsurge in the Bacteroidetes, Firmicutes, and Fusobacteria phyla in their guts [37].

### 3.2. Organochlorine Pesticides (OCPs)

OCPs have been frequently identified in the environment even though several nations prohibited them in the 1970s and 1980s due to their chronic build-up in the body and hazard to human health [47]. The presence of methanobacteriales in the gut, serum OCP concentrations, and obesity in the general population may all be connected, intriguingly. The most well-known and harmful OCPs are p,p’-dichlorodiphenyldichloroethylene (p,p’-DDE) and b-hexachlorocyclohexane (b-HCH), which are the primary breakdown products metabolized from dichlorodiphenyltrichloroethane (DDT) and hexachlorocyclohexane (HCH) [48]. Changes in the gut microbiota composition of mice exposed to DDE for eight weeks increased Bacteroidetes and decreased Proteobacteria, Deferribacteres, and Cyanobacteria [49]. Another analysis revealed that chronic HCH and DDT exposure in mice led to a modification of the relative abundance and stoichiometry of gut microbiota by boosting Bacteroidetes and lowering Proteobacteria, Deferribacteria, and Cyanobacteria, with an increased quantity of the *Lactobacillus* strain being observed with bile salt hydrolase activity (Table 2). These variations substantially impact the hydrophobicity of hepatic and bile acid.

### 3.3. Permethrin (PEM) 

PEM is a pyrethroid compound frequently employed in agricultural, public health, and residential pest management. PEM exposure in the gut microbiota may result from contaminated food [56]. Previous research has found that following exposure to low-dose PEM, the numbers of *Bacteroides*, *Prevotella,* and *Porphyromonas* were decreased. At the same time, the quantity of *Enterobacteriaceae* and *Lactobacillus* in the feces rose after the 4-month follow-up [57]. PEM inhibited both potential pathogens like *Staphylococcus aureus* and *Escherichia coli* and helpful bacteria like *Bifidobacterium* and *Lactobacillus paracasei* in an in vitro investigation [57]. Overall, postnatal exposure to PEM at low doses had a deleterious impact on the gut microbiota in rats, which may play a significant role in the development of illnesses; hence, more research is needed. In conclusion, pesticides have detrimental impacts on the gut microbiota in several animal models and have an effect on the host’s health. Research that lends credence to the theory found that aldicarb exposure enhanced the pathogenicity of gut bacteria in mice and disrupted their metabolism [40]. As a result, researchers have begun paying closer attention to gut bacteria when assessing the toxicity of pesticides.

## 4. Effects of Fungicides on the Gut Microbiota

Fungicides focus on protecting crops from spores or parasitic fungi. Fungicides are widely utilized to prevent rot in vegetables, fruits, or post-harvest crops. Because fungicide residues may enter into water, air, and soil, they have been detected in foods intended for human consumption, most commonly in from post-harvest treatments [39]. The gut microbiota of mammals or zebrafish can be affected in several ways by fungicides, and these alterations can lead to host physiological problems, according to an increasing number of studies over the last few years.

### 4.1. Carbendazim (CBZ)

A fungicide called CBZ (methyl 2-benzimidazole carbamate) prevents microbial growth during agricultural storage and industrial processing [58]. However, CBZ inhibits liver oxidative stress, which leads to reproductive damage, endocrine disruption, and potentiates disease pathogenesis [59]. When CBZ builds up in the GI tract, it interacts with the gut microbiota, causing dysbiosis and thereby reducing the amount of *Bacteroides* and increasing the population of Firmicutes, Proteobacteria, and Actinobacteria [37]. Due to CBZ’s build-up in liver and fat tissue over 28 days after oral high-dose dosing, mice developed hepatic lipid metabolic abnormality, ultimately leading to inflammation [60]. Additionally, continuous exposure of mice to CBZ for 14 weeks caused dysbiosis of the gut microbiota, which led to dyslipidemia. This made it easier for triglycerides to be absorbed through the intestinal wall, which triggered an inflammatory response [61]. As a result, exposure to CBZ may influence the gut microbiota and general health by interfering with the host’s hepatic lipid metabolism and inflammatory response.

### 4.2. Imazalil (IMZ)

The well-known fungicide IMZ is frequently found in soil, water, fruits, and vegetables. Like other fungicides, it protects post-harvest crop products [62]. Developmental toxicity and aberrant locomotor behavior may have been caused by early IMZ exposure in zebrafish development [63]. Dysbiosis and altered hepatic metabolism also occurred when zebrafish were exposed to IMZ for 21 days [64]. Additionally, by lowering the proportion of Firmicutes and boosting Proteobacteria, Actinobacteria, and Bacteroidetes over the course of 28 days, mice’s exposure to IMZ (100 mg/kg body weight/day) may have triggered mouse colonic inflammation and dysbiosis in the gut. IMZ exposure simultaneously decreased the population of *Lactobacillus* and *Bifidobacterium* while increasing the levels of Deltaproteobacteria and Desulfovibrio [56]. Mice given modest doses of IMZ repeatedly showed impaired intestinal barrier function. These findings also show that cystic fibrosis transmembrane conductance regulator expression is downregulated in the ileum and colon of mice exposed to IMZ, along with decreased mucosal release [37].

### 4.3. Propamocarb (PM)

PM, a systemic carbamate fungicide, is employed to manage ailments in soil, roots, and leaves brought on by oomycetes. High quantities of PM residues may build up on fruit, impacting people [47]. High dosages of PM exposure for a brief period were shown to affect the hepatic lipid metabolism in zebrafish. By altering the gut microbiota and microbial metabolites, PM exposure has the potential to disrupt metabolism, according to mounting data. For instance, exposure to 300 mg/L PM for 28 days alters 20 different types of fecal metabolites, including succinate, SCFAs, bile acids, and trimethylamine, in addition to altering the gut microbiota at the phylum or genus level (TMA) [65]. These metabolites also have an impact on the host’s health. Dietary choline can be converted to TMA by the action of the gut microbiota and then further oxidized to TMAO in the liver by flavin mono-oxygenase 3 (FMO3), which is markedly increased by bile acids and is mediated by the farnesoid X receptor (FXR) [47]. Atherosclerosis development and TMAO concentration are tightly correlated. Mice exposed to modest doses of PM for extended periods also showed a substantial shift in the makeup of their gut flora [51]. The compounds in feces that are important in energy metabolism were also changed, drastically changing energy metabolism. TMA, which causes atherosclerosis, was also found in the feces, disrupting the cardiac NO/NOS pathway and increasing NF-kB transcriptional levels. Overall, prolonged PM exposure disrupts enterohepatic metabolism and may raise the risk of cardiovascular disease [51].

### 4.4. Epoxiconazole

Wheat is frequently treated with epoxiconazole, a broad-spectrum azole fungicide, to ward off diseases such as leaf blotch (*Septoria tritici*) and rust *(Puccinia triticina*) [52]. There is no question that eating food contaminated with epoxiconazole increases the likelihood that a person may become exposed. An earlier investigation showed that exposure to epoxiconazole could also alter a rats’ intestinal microbiota makeup [47]. A prior study found that epoxiconazole might alter the gut microbiota structure in rats. Firmicutes abundance dropped while Bacteroidetes and Proteobacteria abundance rose in female rats treated with epoxiconazole (4 and 100 mg/kg/day) for 90 days. Lachnospiraceae and Enterobacteriaceae were preferentially enriched at the family level [66]. Lactobacillaceae have the ability to protect the liver from injury. Bacteroidaceae abundance is correlated with pouchitis, and increasing abundance is associated with Crohn’s disease [52]. According to this study, epoxiconazole exposure may alter the gut flora and may be hazardous to the liver. The findings strongly imply that alterations in the gut microbiota brought on by epoxiconazole may serve as early warning signs for assessing the host’s health risk [47]. In conclusion, fungicides are strong chemicals that have an impact on human health by modifying the microbiota in the gut. However, further research based on mechanisms would be required to meet the substantial need for and the range of research on the direct effects of fungicides on gut flora.

## 5. Effects of Herbicides on the Gut Microbiota

Herbicides are commonly employed as weed killers; however, their residue on food grains such as maize/soya beans may influence non-target species such as people, producing severe health consequences [67]. Insecticides and herbicides have also been found in cereals, fruits, and vegetables used for human consumption. Other crucial human exposure sources to these chemicals include drinking water, air, and soil [68]. Several studies have recently been conducted to examine the influence of these chemicals on gut microbiota and disease incidence in diverse animal models.

### 5.1. Glyphosate (GLY)

Glyphosate (GLY) is the most widely used herbicide for eradicating plants and grasses. Glyphosate exposure has been related to deleterious consequences in animals via a variety of methods [69]. Adding three aromatic amino acids (tyrosine, tryptophan, and phenylalanine) attenuated the inhibitory impact of glyphosate in a dose-dependent manner, according to prior research [70]. The dietary intake of appropriate amounts of aromatic amino acids reduces the demand for the bacterial manufacture of these compounds, preventing glyphosate from having an in vivo antimicrobial impact [70]. However, the adverse effects of glyphosate cannot be completely ruled out in instances of human malnutrition or people following particular diets (such as low-protein ones), as that may result in reduced quantities of amino acids being readily accessible in the stomach. Recently, several scientists have studied the impacts of glyphosate on gut microbiota using various animal models [71]. The gut–brain axis may be negatively impacted by ambient glyphosate levels, which may increase the development of *Clostridium* bacteria in autistic children [72]. According to a recent study, Hawaiian green turtles exposed to glyphosate at a concentration of 2.2 × 10^−4^ gL^−1^ had significantly lower bacterial densities than controls. Additionally, glyphosate drastically reduced the number of *Proteus, Shigella, Pantoea*, and *Staphylococcus* in a dose-dependent way, suggesting that glyphosate may negatively impact gut microbiota and general health [4]. Rats exposed to glyphosate saw changes in the villi’s morphology in their duodenum and jejunum. Additionally, exposure to glyphosate increased inflammatory responses by upregulating the expression of genes involved in IL-1, IL-6, TNF-, caspase-3, Mapk3, and NF-B. This reduced the antioxidant system. Rats exposed to glyphosate had a considerably lower abundance of Firmicutes than other phyla [73]. Some researchers examined how GLY affected the microbiota of the mussel “*M. galloprovincialis*”, and they hypothesized that the altered bacterial species’ proliferative patterns following exposure to GLY might result in microbiota dysbiosis, which would then encourage the spread of opportunistic pathogens [72].

GLY treatment, either alone or in a formulation, impacted the rats’ GM; an accumulation of shikimic acid and an increased level of metabolites were discovered in the cecum but not in the serum, indicating that the effect was limited to the gut microbiota [73]. Reduced digestive enzyme activity and a variety of microbiota were the results of the glyphosate-based herbicide’s (Roundup) effects on the immune system and gut microbiota of the crab “*Eriocheir sinensis*”, but increased amounts of Bacteroidetes and Proteobacteria were seen at the taxonomic level [40].

### 5.2. Pentachlorophenol (PCP)

Widespread usage of PCP as a herbicide for weeding rice fields has resulted in significant PCP pollution of the aquatic environment. It is interesting to note that research on goldfish has shown how the harmful effects of PCP changed the gut microbial flora [74]. Additionally, fish exposed to PCP for 28 days showed growth suppression, increased oxidative stress, and histopathological damage due to PCP’s propensity to accumulate in the liver and gut [47]. Additionally, it led to a change in the fish gut’s Bacteroidetes/Firmicutes ratio, which changed the microbial ecology (Figure 2). These results have sparked more attention to the effects of PCP-induced toxicity on host health and dysbiosis of the gut microbiota [37]. Furthermore, through gut microbiota dysbiosis, antibiotics have dramatically increased the hazards associated with triazine use, supporting a potential interaction between antibiotics, herbicides, and triazine. In the fish gut, the quantities of Chryseobacterium, Microbacterium, Arthrobacter, and Legionella were reduced, which may have contributed to the rise in Bacteroidetes [42].

## *6.* Effects of Pesticides on Gut Microbiota-Mediated Host Immune System

The critical gut microbiota has been well-recognized to play a vital role in regulating the signaling pathways involved in the optimal functioning of the host immune system and maintaining homeostasis [42]. As expected, any imbalance or deviation from the optimal microbial composition or function can profoundly impact the immune system, increasing the susceptibility to infections and leading to the development of inflammations. In this regard, exposure to chemicals including pesticides can have deleterious effects on the gut microbiota-mediated immunity of the host, owing to dysbiosis, increased permeability, and inflammation, further extending serious implications on homeostasis and host metabolism.

Intestinal epithelial cells act as an essential barrier in delimiting the invasion of host tissues by bacteria and other microbes and coordinating the functions of the subepithelial immune cells. Pesticides have been implicated in disrupting the epithelial barriers of the host, possibly by the disruption of tight junctions, such as claudins, and adherent junctions, like cadherins, in a ROS-dependent manner, thereby leading to increased permeability of the epithelial barrier [75]. For example, in an early study conducted by Tirelli et al., chlorpyrifos was shown to affect the epithelial barrier, as demonstrated using colon tissue-derived Caco2 cell lines, which showed a marked decrease in the expression of tight junction molecules [76]. Furthermore, given the effects of pesticides on the host’s health, such disruption of the epithelial barrier leads to gut microbiome dysbiosis. Such dysbiosis in the gut can further lead to intestinal inflammation and disruption of the host’s immune function by triggering innate immune responses on the host epithelium and other immune cells including macrophages, mast cells, keratinocytes, etc., further driving systemic proinflammatory responses [77]. In another study conducted by Zhu et al., 2020, the effects of nitenpyram, an insecticide, on gut microbiota dysbiosis and related immunity impairments were studied on Apis mellifera. Upon exposure to the compound, a marked reduction in the abundance of Gammaproteobacteria was observed, along with an increase in the abundance of Betaproteobacteria and Alphaproteobacteria, indicating significant dysbiosis. Interestingly, a concurrent alteration in the genes related to host immunity and pathogen defense was also observed, including a downregulation of genes involved in the expression of the bee antimicrobial peptide escaping. Furthermore, a significant inhibition of toll-like receptors (TLRs) and other defense protein-encoding genes was also observed. Such studies confirm a correlation between host exposure to pesticides and the impact of pesticides on gut microbiota-mediated host immunity [78]. In another study conducted on mice models, the impact of chlorpyrifos on the gut microbiota, the expression of pro-inflammatory cytokines, and the impact on insulin resistance and obesity were analyzed. It was found that exposure to chlorpyrifos resulted in increased gut epithelium permeability by reducing the expression of tight junction proteins and causing a resultant low-grade inflammation through the expression of pro-inflammatory factors such as TNF and TLRs due to increased levels of LPS in the plasma. These findings were concurrent with an altered composition of the gut microbiome, mainly exhibiting an increase in Bacteroidetes and a dencrease in Proteobacteria, confirming the considerable impact of pesticides on overall gut microbial composition and microbiota-mediated host immune function and further resulting in insulin resistance and obesity [50]. 

## 7. Other Potential Environmental Contaminants

Although heavy metals are one of the environmental pollutants thoroughly investigated, it is unknown how they contribute to gut toxicity. By enhancing or lessening the harmful effects of heavy metals, the gut bacterial flora plays a crucial role in the biotransformation of these pollutants [79]. For instance, exposure to arsenic is one of the primary risks to human health from drinking polluted water. According to research, human gut bacteria can transform inorganic arsenic into a less harmful organic form [80]. Additionally, arsenic exposure in the Bangladeshi population has significantly altered the gut microbiota by encouraging an increase in Citrobacter, which is linked to several human health issues, such as urinary tract infections, respiratory conditions, an inflamed digestive system, and cardiovascular disease, specifically atherosclerosis through TMAO [81]. Mice’s gut metabolic profiles changed after exposure to arsenic, cadmium, cobalt, chromium, and nickel in drinking water. Another piece of research corroborated this conclusion by demonstrating the change in gut flora after exposure to 100 ppb of arsenic in mice134, opening up new opportunities to examine the mechanisms behind arsenic-induced harmful consequences in humans [73]. Prior research suggests that exposure to heavy metals such as arsenic, cadmium, lead, and manganese may contribute to the inflammatory processes in the gut through a reduction in indole lactic acid, daidzein, and glycocholic acid, as well as vitamin E increasing LPS (Lipopolysaccharide) [80]. Other animals that have been well investigated regarding heavy metals impact on their microbiota are fish, mice, rats, chickens, and humans. Mercury has also been researched in great detail in these models. Methylmercury exposure damages the GI tract and changes the microbiota in the gut [79]. Nonetheless, there is a need for more research that investigates the mechanism behind heavy metal-induced toxicity of the gut microbiota. More mechanistic investigations of the harmful effects of heavy metals on the gut microbiota and metabolic change are urgently needed in the future.

## 8. Future Perspectives

Given the detrimental effects of pesticides on human health, particularly regarding the gut microbiota, it is critical to expand the avenues of our understanding of long-term exposure and its consequences on the human gut microbiome. Pesticide exposure has been linked in all known research to changes in host gut microbiota composition and metabolic characteristics. Investigations into the mechanisms behind altered microbiome composition and functionality are essential in understanding such chemicals’ true impact on human health. Another interesting avenue in this regard is unraveling the interactions between the gut microbiota and the immune responses of the host in terms of pesticide exposure. This could further highlight the mechanisms behind the modulation of the gut community by commonly employed widespread pesticides and their impact on other immune system-related functions and diseases. Furthermore, research has yet to demonstrate a precise association between a specific pesticide and microbiota or metabolite to determine its toxicity. As a result, there is plenty of scope to investigate the mechanisms behind pesticide toxicity in the gut microbiota and related metabolites. Furthermore, comprehensive assessments of the collective impact of two or more pesticides could highlight the complexities of pesticide-induced microbial disruptions and microbial dynamics. Given the developing effective interventions, biopesticides might be a potential solution to prevent the adverse effects of pesticides on human health. They might comprise pathogen-targeting microorganisms or RNA-based biopesticides. Alternatively, exploring the role of probiotics or prebiotics in mitigating the adverse effects of pesticides holds good potential in identifying certain microbial strains that can be used to restore the desirable resident microbiota in cases of pesticide exposure. Other potential approaches include personalized medicine and synthetic biology, which may provide a better future for agriculture while causing minimum harm to human health. Molecular approaches involving shotgun sequencing and other more in-depth sequencing methods, along with robust omics approaches including proteomics and metabolomics, should be used in the future to identify the strain and species, which are expressly changed owing to intestinal exposure to pesticides. These technologies are also expected to provide a comprehensive view of the functional and metabolic alterations induced in the host gut microbiome, along with an elaboration of a “cause and effect” relation between pesticide exposure and gut microbiota-mediated host functions.

## 9. Conclusions

The intricate relationship between pesticides and the gut microbiota unveils unintended consequences, impacting both human health and the environment. Pesticides disrupt microbial diversity and functionality, influencing immune function, metabolism, and disease susceptibility. Different pesticide classes exert distinct effects on the gut microbiota, emphasizing the need for nuanced risk assessments. The reciprocal relationship, wherein microbial composition affects pesticide metabolism, adds complexity. Recognizing these consequences is vital for shaping sustainable agricultural practices that balance environmental health and human well-being. This review provides insights for future research and policy development in pesticide use and regulation.

## Figures and Tables

**Figure 1 metabolites-14-00155-f001:**
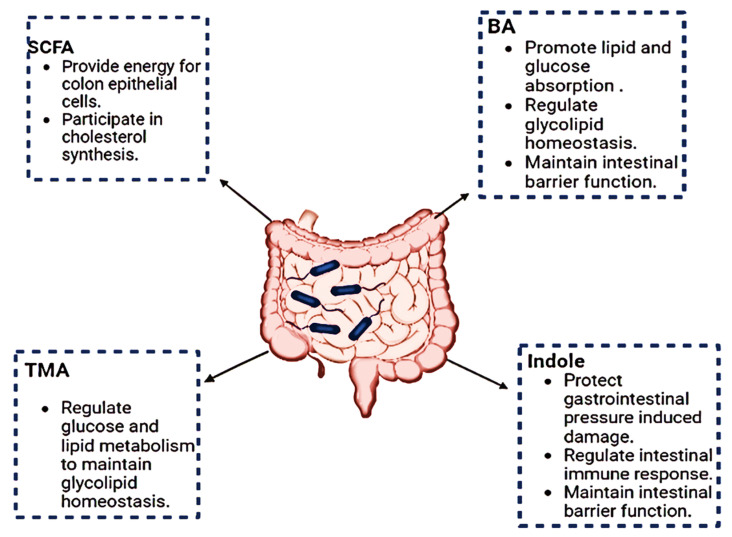
The function of metabolites synthesized by gut microbes. SCFA: short-chain fatty acids; BA: bile Acids; TMA: trimethylamine.

**Figure 2 metabolites-14-00155-f002:**
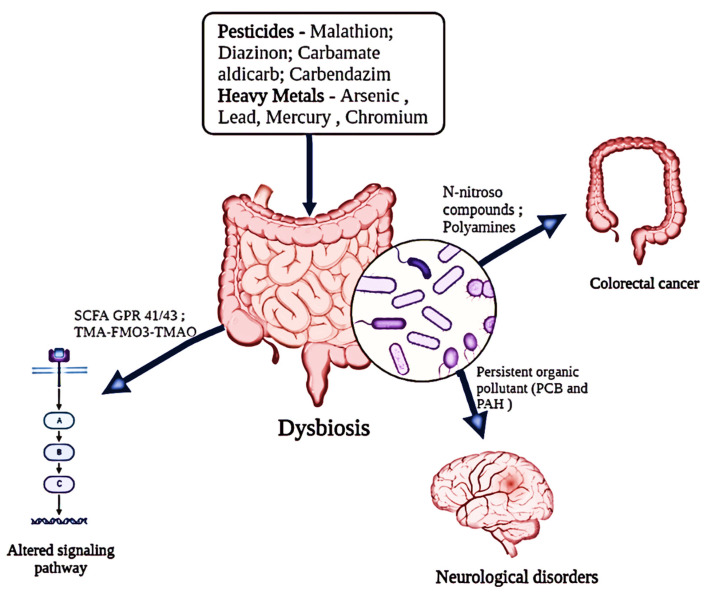
Impact of diverse environmental pollutants (pesticides and heavy metals) on the functional alterations of the host’s gut microbiota.

**Table 1 metabolites-14-00155-t001:** List of major bacterial phyla and species in human gut microbiota.

Phyla	Species	References
Bacteriodetes	*Bacteriodes*, *Prevoetella* and *Xlanibacter*	[12]
Firmicutes	*Ruminococcus*, *Clostridium*; *Lactobacillus* and *Enterococcus*	[13]
Actinobacteria	*Bifidobacterium*	[14]
Proteobacteria	*Escherichia* and *Enterobacteriaceae*	[15]
Verrucomicrobia	*Akkermansia muciniphila*	[16]

**Table 2 metabolites-14-00155-t002:** Effect of the different compounds on gut microbiota.

Compound	Impact on Gut Microbes	References
Chlorpyrifos	Increased Proteobacteria population, decreased Bacteroidetes; significant impact on insulin resistance and obesity in test mice	[50]
Organochloride pesticides (p,p’-dichlorodiphenyldichloroethylene; β-hexachlorocyclohexane)	Significant alterations in composition of gut microbiome; extended effects on bile acid metabolism, with potential effects on human health	[51]
Organophosphates, glyphosate, pyrethroid	Altered composition in gut microbiota, significantly among Firmicutes, Bacteroidetes, and Lactobacilli	[52]
Pyrethroid (bifenthrin)	Dysbiosis in gut microbiota upon exposure to pesticides, significant impact on lipid metabolism, and development of non-alcoholic fatty liver disease in *Xenopus laevis*	[53]
Pyrethroid (deltamethrin, cypermethrin, and permethrin)	Altered gut microbiome in Aedes albopticus model, with decreased impact on Butyricimonas, Prevotellaceae, Anaerococcus, Pseudorhodobacter and enriched occurrence of *Wolbachia, Chryseobacterium, and Pseudomonas*	[54]
Carbendazim	Significant decrease in richness and diversity of gut microbiota, and observable decrease in populations of Bacteroidetes and an increase in Firmicutes, Proteobacteria, and Actinobacteria in mice models exposed to carbendazim, along with an simultaneous occurrence of lipid metabolism disorders in the mice	[55]

## Data Availability

Not applicable.

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
