# Peer review of "Pesticides: Unintended Impact on the Hidden World of Gut Microbiota"

_metabolites, 2024, doi:10.3390/metabo14030155_

Round 1

Reviewer 1 Report

Comments and Suggestions for Authors

This review summarized recent findings to elucidate a potential correlation between pesticides and disrupted gut microbiota, highlighting the need for future research. This review provides certain valuable information to its potential audiences. A few minor changes are suggested.

-       I am unsure about the role of heavy metals, also called other potential environmental contaminants, in this review. Is there a possible connection between this and the main topic, namely, the pesticides? Please elucidate.

-       The effects of disrupted gut microbiota can be profound on the human body. There are other health issues that might be induced by pesticides, such as cancer, endocrine alterations, and neurological damage. What do the authors think about the relationship between the altered microbiota and these other health issues?

-       Please double-check the references. Some should be listed in the reference section, such as lines 31, 39, and 214. References 2 and 3 are the same. 

-       The language needs improvement. It would be a good idea to have a native speaker go through and improve the whole draft.

-       The resolution of the figures also needs to be increased.

Comments on the Quality of English Language

The language needs to be further improved. 

Author Response

Response to referees' comments

Manuscript title: “Pesticides: Unintended Impact on the Hidden World of Gut Microbiota”

Manuscript ID: metabolites-2885899

************************************************************************************************

Editor

We express our gratitude to the esteemed editor and reviewers for dedicating their time and effort to provide valuable suggestions for improving the manuscript. We have diligently considered and addressed the comments provided by the reviewers to the best of our ability and with the utmost care.

Reviewer 1:

Query 1:

I am unsure about the role of heavy metals, also called other potential environmental contaminants, in this review. Is there a possible connection between this and the main topic, namely, the pesticides? Please elucidate.

Response:

I would like to express my gratitude for the valuable suggestion provided. The primary focus of the review pertains to pesticides and their impact on gut microbiota; it is pertinent to acknowledge the relevance of considering other potential environmental contaminants, such as heavy metals. Heavy metals, akin to pesticides, can inadvertently influence gut microbiota due to their ubiquitous presence in the environment and their ability to disrupt microbial communities. Moreover, exposure to heavy metals and pesticides frequently transpires concurrently, particularly in agricultural settings where both types of contaminants may coexist in soil and water reservoirs. Consequently, delving into the potential interactions and cumulative effects of these diverse contaminants on gut microbiota health holds promise for yielding valuable insights into the overall environmental repercussions on microbial ecosystems. I appreciate your attention to this aspect.

Query 2:

The effects of disrupted gut microbiota can be profound on the human body. There are other health issues that might be induced by pesticides, such as cancer, endocrine alterations, and neurological damage. What do the authors think about the relationship between the altered microbiota and these other health issues?

Response:

Thank you for drawing attention to this crucial aspect. Indeed, disturbances in gut microbiota can have profound implications for human health that extend beyond the scope of our current review. While our review focuses specifically on the effects of pesticides on gut microbiota, it is imperative to acknowledge the potential correlations between altered microbiota and other health concerns induced by pesticides, such as cancer, endocrine disruptions, and neurological impairments. Although a brief mention of gut-associated diseases is made in section 2.2, recent studies indicate that disruptions in gut microbiota composition and function may contribute to the onset or exacerbation of various health conditions, including those previously mentioned. For instance, specific alterations in gut microbiota composition have been linked to heightened inflammation, oxidative stress, and metabolic dysregulation, all recognized as risk factors for cancer and other chronic ailments. Furthermore, the gut-brain axis, facilitating bidirectional communication between the gut and the brain, could potentially mediate pesticide-induced neurological damage by modulating neurotransmitter production and neuroinflammatory responses. While our review primarily addresses the direct effects of pesticides on gut microbiota, it is crucial to acknowledge the potential indirect impacts on overall health through alterations in microbial communities. We wish to inform the reviewer respectfully that a particular aspect of the influence of altered gut microbiota on human health has been previously addressed in another review by our team, albeit not in relation to pesticides specifically. Your highlighting of this point is greatly appreciated.

Query 3:

Please double-check the references. Some should be listed in the reference section, such as lines 31, 39, and 214. References 2 and 3 are the same.

Response:

The authors express their gratitude to the reviewer for identifying these errors in the manuscript. The references have been appropriately addressed.

Query 4:

The language needs improvement. It would be a good idea to have a native speaker go through and improve the whole draft.

Response:

The language throughout the manuscript has been refined in accordance with the suggestions provided by the reviewer.

Query 5:

The resolution of the figures also needs to be increased.

Response:

The authors thank the reviewer for drawing attention to this matter. All figures in the manuscript have been substituted with images of higher resolution.

Reviewer 2 Report

Comments and Suggestions for Authors

The authors of the manuscript describe the results of a literary review of 89 studies that reported on the relationship between the gut microbiome, its metabolic activity and the effect of insecticides, herbicides and fungicides on the intestinal microbiota.

The review was carried out at a good methodological level. Literary references are mostly represented by contemporary works. The data and conclusions reached by the authors of the manuscript may be of interest to researchers in the field of pediatrics, neurobiology and nutritionology, etc. Therefore, the conducted research will be relevant for publication in the journal Metabolites.

However, I recommend that Section 2 be significantly shortened, since these data only indirectly relate to the subject of the manuscript.

Author Response

Response to referees' comments

Manuscript title: “Pesticides: Unintended Impact on the Hidden World of Gut Microbiota”

Manuscript ID: metabolites-2885899

************************************************************************************************

Editor

We express our gratitude to the esteemed editor and reviewers for dedicating their time and effort to provide valuable suggestions for improving the manuscript. We have diligently considered and addressed the comments provided by the reviewers to the best of our ability and with the utmost care.

Reviewer 2:

Query 1:     

I recommend that Section 2 be significantly shortened, since these data only indirectly relate to the subject of the manuscript.

Response:

Thank you for your recommendation regarding Section 2. We agree that streamlining this section is advisable, and appropriate modifications have been integrated into this specific section of the revised edition of the manuscript.

Reviewer 3 Report

Comments and Suggestions for Authors

The manuscript “Pesticides: Unintended Impact on the Hidden World of Gut Microbiota” is an interesting review of the possible effects of different pesticides (insecticides, fungicides, herbicides) and other contaminants such as heavy metals, on the intestinal microbiota.

Text is relatively easy to read (despite the hardness of the technical information on the different compounds it analyzes) and I think it is a good update on the topic.

In my opinion it could be published in Metabolites with minimal changes that I now point out:

1.- I believe that the Figures should be improved in quality to be published and should carry the meaning of the acronyms they use. Figures and tables must be understandable on their own.

2.- I think point 5.3 should be independent, since it analyzes the role of pesticides in general (and not only herbicides) on the immune system. Could be a new point "6".

3.- Point 6 (effect of other contaminants such as heavy metals) should be justified to maintain its inclusion in the article.

4.- It could be interesting to introduce a new table where the effects of the different compounds are summarized in some way with very short information.

Comments on the Quality of English Language

Moderate editing of English language required.

Author Response

Response to referees' comments

Manuscript title: “Pesticides: Unintended Impact on the Hidden World of Gut Microbiota”

Manuscript ID: metabolites-2885899

************************************************************************************************

Editor

We express our gratitude to the esteemed editor and reviewers for dedicating their time and effort to provide valuable suggestions for improving the manuscript. We have diligently considered and addressed the comments provided by the reviewers to the best of our ability and with the utmost care.

Reviewer 3:

Query 1:     

I believe that the Figures should be improved in quality to be published and should carry the meaning of the acronyms they use. Figures and tables must be understandable on their own.

Response:

We would like to express our gratitude for bringing the irregularities to our attention. Desired changes in figure captions have been included in the revised version.

Query 2:     

I think point 5.3 should be independent, since it analyzes the role of pesticides in general (and not only herbicides) on the immune system. Could be a new point "6"..

Response:

As per the reviewer’s suggestion, the topic addresses a distinct aspect, and consequently, it has been incorporated into the manuscript as a new subheading under point "6".

Query 3:     

Point 6 (effect of other contaminants such as heavy metals) should be justified to maintain its inclusion in the article.

Response:

The rationale for incorporating this point stems from the interrelation of environmental contaminants and their potential cumulative impacts on gut microbiota. Real-world scenarios frequently entail simultaneous exposure to multiple contaminants, particularly in agricultural contexts. Furthermore, studies suggest that concurrent exposure to substances such as heavy metals and pesticides may result in synergistic or additive effects on gut microbiota, intensifying disturbances in microbial ecosystems. Diving into these broader ecological dynamics fosters a comprehensive comprehension of the environmental pressures confronting gut microbiota and emphasizes the significance of addressing the effects of diverse contaminants beyond pesticides exclusively.

Query 4:     

It could be interesting to introduce a new table where the effects of the different compounds are summarized in some way with very short information.

Response:

Thank you for the observations provided. In accordance with your suggestion, we have summarized the information in Table 2.
